# Players’ Physical Performance Decreased After Two-Thirds of the Season: Results of 3 Consecutive Seasons in the German First Bundesliga

**DOI:** 10.3390/ijerph16112044

**Published:** 2019-06-10

**Authors:** Paweł Chmura, Marek Konefał, Del P. Wong, Antonio J. Figueiredo, Edward Kowalczuk, Andrzej Rokita, Jan Chmura, Marcin Andrzejewski

**Affiliations:** 1Department of Team Games, University School of Physical Education, 51-612 Wrocław, Poland; pawel.chmura@awf.wroc.pl (P.C.); andrzej.rokita@awf.wroc.pl (A.R.); 2Department of Biological and Motor Sport Bases, University School of Physical Education, 51-612 Wrocław, Poland; jan.chmura@awf.wroc.pl; 3Titi Sport Technology Company, Shenzhen 510810, China; delwong@126.com; 4Faculty of Sport Sciences and Physical Education, University of Coimbra, 3040-156 Coimbra, Portugal; afigueiredo@fcdef.uc.pt; 5Football Club, Hannover 96, 30169 Hannover, Germany; edward.kowalczuk@hannover96.de; 6Department of Recreation, University School of Physical Education, 60-101 Poznań, Poland; andrzejewski@awf.poznan.pl

**Keywords:** soccer, motion analysis, match running performance, phases, intra-seasonal variation

## Abstract

The study aimed to: (1) investigate physical and technical performance among players during six consecutive phases of a competitive season, (2) determine if levels of match running and technical performance between the 1st and 6th phases of a season can be maintained, (3) and determine which phase features the lowest and highest values for variables assessed. Seventy out of 556 outfield players who played at least 80% of the matches in each phase of the Bundesliga seasons were analysed. Each season was divided into 6 phases: phase 1 (matches 1–6), phase 2 (7–11), phase 3 (12–17), phase 4 (18–23), phase 5 (24–28) and phase 6 (29–34). Thirteen variables were selected to quantify the physical and technical activity of players. Our results showed that by reducing the distances covered at intensities below 11 km·h^−1^, players were able to cover greater distances at intensities in the range of 11–13.99 and 21–23.99 km·h^−1^ in subsequent phases of a season. Players’ capacity to maintain match running and technical performance between the first and sixth phases of the season has been demonstrated, though the 4th phase of the season showed a breakthrough point as regards the maintenance of exercise capacity among players participating in the Bundesliga.

## 1. Introduction

In contemporary European professional soccer, the fact that a competitive season lasts 9–10 months ensures that teams can participate in over 60 competitive matches [1,2,3,4]. It therefore becomes rather difficult for players to maintain optimal physical fitness throughout a season, even if this is a target for all teams at the outset. It is further acknowledged that ‘natural’ variations in physical activity profiles occur over consecutive games during different stages of the competitive season [5,6]. The in-season training programme can be challenging for coaches, since they are required to prescribe training loads that maximise positive physiological adaptation, while avoiding overtraining and injury among players [2,7]. Proper adjustment of the total training stress both over the long-term and within short-term training cycles is considered essential to success [8]. However, the periods over which players are genuinely exposed to a risk of decline in ensuing match running performance is still unclear [9].

Elite soccer players compete once or even twice a week, so decisions on players’ wellness and fatigue are frequently required over extended periods of time [2]. Hence, there is potential for professional soccer players to experience transient/residual fatigue over the playing season, especially when competitive schedules are congested [3,10]. Fatigue following a soccer match is multifactorial and mainly related to dehydration, glycogen depletion, intramuscular acidosis, central and psychological fatigue [11,12,13]. Despite a more systematic approach to physical conditioning, statistically significant declines in running activities are apparent, indicating a susceptibility to fatigue and an inability to perform consistently over the course of play, even at the highest standards of contemporary competition [5,14,15,16]. 

The Bundesliga is one of the most influential leagues in the world, given the relevant international results obtained by Germany’s league and national teams [17,18]. Analysis of the specificities of this league would be relevant, but to the best of our knowledge there has been limited research to date on the effects of numbers of matches played over subsequent phases of a season on physical and technical performance [2,14,16]. Most research results available reflect only a part of a season, a short period of time, or a small number of players [7,8]. Thus far, comparisons have at best examined variability between matches or between two rounds of the season [5,14,19,20].

There remain many aspects in need of analysis if the character of the game is to be understood better. One such aspect is physical and technical performance in subsequent phases of a season [14,21]. This aspect needs to be well recognised, with the selection from among all players in a league confined to those playing the greatest number of full matches during a season. Studies need to further analyse the variability of performance in the same group of players during different periods of a season [22]. Multiple longitudinal observations over an extended period of time are therefore required for a more comprehensive assessment of competitive performance indices [1]. By examining changes in players’ physical activity throughout a season, insight into the impact of number of matches played on performance in team sports would be gained [16,23]. 

Many authors have long pointed to the need to examine peak periods, so that the worst-case scenarios players encountered can be taken into consideration. The identification of these periods should prove more informative for practitioners who prepare players for these scenarios [24]. In their recent research, Ryan, Coutts, Hocking and Kempton [21] indicated that phase of the season should be considered by coaches and high-performance staff as they interpret features of a physical nature [5,25,26], and a technical nature [27,28,29], among players during an entire season. The division of a season into more phases allows for a more precise determination of directions and a better understanding of changes in players’ match running performance and technical activities. Knowledge as to phases of the season and the parameters worthy of special attention can help to counteract the negative effects of increasing fatigue in consecutive league matches (such as players’ reduced technical and tactical precision of action, followed by the severity of errors and reduced involvement in the game).

For the abovementioned reasons, this study aimed to: (1) investigate physical and technical aspects of players’ performance in six consecutive phases of a league season, (2) determine if there are soccer players who can maintain a level of match running performance and technical activity throughout the six phases of a season and (3) determine which phases feature the lowest and highest peak periods for physical and technical performance. We hypothesised that: (1) players’ match running and technical performance changes between consecutive phases of the season, (2) the length of the season and large number of matches played may cause a decrease in physical and technical performance and (3) there are peak periods in the season in which players manifest their highest levels of performance.

## 2. Materials and Methods 

### 2.1. Experimental Approach to the Problem

An objective analysis of physical and technical activities engaged in by professional soccer players was carried out using the IMPIRE AG match system deployed widely in German football. In this longitudinal study, three seasons of the German Bundesliga were analysed, i.e., the 2014/2015, 2015/2016 and 2016/2017 seasons. Each season was divided into 6 phases to specify figures for the variables assessed: phase 1 (matches 1–6), phase 2 (matches 7–11), phase 3 (matches 12–17), phase 4 (matches 18–23), phase 5 (matches 24–28), and phase 6 (matches 29–34).

### 2.2. Players and Match Data

Match performance data were collected from 70 out of 556 outfield players, excluding goalkeepers. Mean body height among players was 183.92 ± 7.12 cm, mean body mass was 78.57 ± 7.34 kg and mean age was 26.64 ± 4.03 years. Analysis was confined to players playing at least 80% of the matches in each phase of a season and playing for the entire durations of matches. The difficulty of selecting players for longitudinal study is highlighted by the fact that only 3 players participated in 100% of matches in one of the three analysed seasons. If a player was not present in a given match, the value of the examined physical and technical performance for that player was assumed to be the arithmetic mean computed for the other players playing in the same phase. Injured players were excluded from the analysis. The study was conducted in compliance with the Declaration of Helsinki and was approved by the local ethics committee (No. 20/2017). The study protocol was also approved by the Board of Ethics of the University School of Physical Education in Wrocław, Poland.

### 2.3. Data Collection and Analyses

The analysis was carried out using an IMPIRE AG motion analysis system [18], providing records of all players’ movements in all 918 matches, with a sampling frequency of 25 Hz. IMPIRE AG (Ismaning, Germany) and Cairos Technologies AG (Karlsbad, Germany) provide a ready-to-use vision-based tracking system for team sports called VIS.TRACK. That system consists of two cameras and offers software tracking of both players and the ball [30]. The major advantages of vision-based systems lies in their high update rate corresponding to the camera frame rate, and how they simultaneously track players and the ball, i.e., each position sample for a single player has a corresponding position sample for every other player including the ball, measured at the identical point in time [31]. The validity and reliability of this system for taking such measurements have been described in detail elsewhere [18,30]. Furthermore, Liu et al. [32] showed that team match events coded by independent operators using this system achieved very good levels of agreement (weighted kappa values of 0.92 and 0.94), with the average difference of event time equal to 0.06 ± 0.04 s. The recorded variables included selected physical activities of players such as total distance covered [km], and distances covered [km] at intensity ranges of below 11 km·h^−1^ (standing intensity—standing, walking, jogging), 11–13.99 km·h^−1^ (low intensity), 14–16.99 km·h^−1^, (moderate intensity), 17–20.99 km·h^−1^, (high intensity) 21–23.99 km·h^−1^ (very high intensity) and above 24 km·h^−1^ (sprinting) [15,20,25,26,33], as well as selected technical activities of players such as ball possession [s], passes [number], passing accuracy [%], 1-on-1 duels [number], wins of 1-on-1 duels [number] and wins of 1-on-1 duels [%] [27,28,29].

### 2.4. Statistical Analysis

All the variables were checked to verify their conformity with a normal distribution. Arithmetic means and standard deviations were calculated. Repeated-measures ANOVA was used to compare mean values for the examined variables. Fisher LSD (Least Significant Difference) post-hoc tests were performed to assess differences between means. All statistical analyses were performed using the Statistica ver. 13.1 software package (Dell Inc., Tulsa, OK, USA). The level of statistical significance was set at *p* ≤ 0.05.

## 3. Results

Data for total distance covered by players in the Bundesliga with regard to phases of the season revealed these soccer players ran a significantly greater distance in the 2nd phase of the season as compared with the 1st (*p* ≤ 0.05, Figure 1). This parameter then stayed at a similar level until the end of the 3rd phase. In the 4th phase, a higher value for the variable was recorded. Subsequently, it was reported that players in the 5th phase ran a significantly shorter total distance than that in the 4th, and a significantly longer total distance as compared with the 6th (*p* ≤ 0.05).

ANOVA results indicated a significant difference in the low-intensity running distance between the 6 phases (F = 21.27, *p* ≤ 0.05). Analysis of distances covered by players at intensities below 11 km·h^−1^ showed that they ran steadily shorter distances in successive phases of a season up to phase 4 (*p* ≤ 0.05). After that, players ran significantly longer distances at this intensity during the 5th to 6th phases of the season (*p* ≤ 0.05) (Table 1).

Consideration of distances covered at intensities of 11–13.99, 14–16.99, 17–20.99 and 21–23.99 km·h^−1^ demonstrated an analogous reverse course for changes. The distances covered by players increased steadily from the 1st to the 4th phases of the season (*p* ≤ 0.05) (Table 1). At all of these intensities, the greatest distance players covered was in the 4th phase. There was then a significant decline in the 5th and 6th phases of the season (*p* ≤ 0.05). Thus, when the 1st and 6th phases were compared, no significant differences in distances travelled were observed, at any intensities of activity ranging from low through to very high.

Finally, where account was taken of distances covered by players at intensities exceeding 24 km·h^−1^, the highest recorded value occurred in the 5th phase of the season, with a significant decrease in values for the tested variable noted subsequently in the 6th phase (*p* ≤ 0.05) (Table 1).

Analysis of players’ technical performance revealed that, leaving aside a significant increase between the 1st and 2nd phases in the time for ball possession and the number of passes they made, all other variables tested manifested similar values in consecutive phases of the season, with no statistically significant differences found (Table 2).

## 4. Discussion

The first aim of the present study was to investigate selected aspects of the physical and technical performance among players in consecutive phases of a soccer league season. The inclusion of the players from multiple teams who played most full matches offered a unique analysis of the effects of consecutive phases of the season on players’ match running performance and technical activity. The major findings of the present study were a gradual decline from the 1st through to the 4th phases in the distance covered by players at an intensity below 11 km·h^−1^, as followed by an increase through to the 6th phase. The opposite direction to these changes was observed by analysing distances covered at intensities ranging from low to very high. Up to the 4th phase of the season, the distances covered by players across this range of intensities increases, only to decrease again subsequently [15,20,25,33]. Furthermore, our research shows that the distances covered by players at intensities exceeding 24 km·h^−1^ remain relatively constant from the 1st to the 5th phases, though the 6th phase brings a significant fall in values for this variable. Previous physiological research has confirmed that professional players’ aerobic fitness increases from pre-season to mid-season before subsequently decreasing. Anaerobic power remains unchanged through a season, whereas agility and sprint performance both deteriorate in the off-season, before improving as a result of pre-season training [34]. Moreover, across a season, changes in performance might be linked to alterations in the physical condition of the players, as the activity profile in match play can fluctuate in conjunction with the amount of fitness training a team completes [35]. However, consecutive high-intensity training and matches during a season lead to mental and physical fatigue [36].

Reported fluctuations in selected aspects of physical performance in consecutive phases of a season did not affect players’ engagement in technical activity, especially at the end of the season. In agreement with this, Carling and Dupont [15] report no significant changes in technical activity through the season among players. Our results can further inform off-season individual training or modification of the training load in the summer pre-season, especially in terms of aerobic endurance. It will help players adapt more rapidly to match effort in first matches, and allow them to outperform opponents in terms of oxygen efficiency in the 1st phase of the season [16]. In addition, by increasing the number of exercises focusing on aerobic power in the 5th and 6th phases [21], maintenance of players’ match running performance will be encouraged, while overloading of the season and risk of injury will be avoided [13].

The second aim was to determine if there are soccer players who can maintain their level of match running performance and technical activity from the 1st to 6th phases of the season. Most current research is consistent with the idea that top-level players are able to maintain match running performance even in congested periods [37,38,39]. In our study, no difference between the beginning and end of the season was found in relation to either physical activity or technical activity. This means that, regardless of match running, players are able to maintain a level of technical activity through the entire season [40]. Dupont, et al. [41] and aus der Funten, et al. [42] found that, among top-level players, rest time between 72 and 96 h after a game was sufficient to ensure full regeneration of the body and a return to the optimal level of physical activity in players. Bloomfield, et al. [43] and Konarski [44] both stated that training confers on players an ability to respond quickly and properly to increasing fatigue levels, allowing them to achieve an optimal level of recovery in advance of the season’s next game. Some attribute this to players employing conscious and/or subconscious pacing strategies and/or self-regulation with a view to physical and technical performance being maintained effectively throughout the latter stages of the season [9,37,45]. Moreover, possible explanations for this include changes in tactical behaviour during the season, or improved physical capacity as an adaptation to competition matches as the season progresses. At this level of competition, players are probably also the subjects of systematic strategies of post-match recovery methods (alternating hot and cold water immersion of legs, massage, diet and drink supplementation) [38,39]. Players may therefore focus on greater efficiency and effectiveness where technical performance is concerned (especially as regards accuracy of action and “key” passes) [27].

The third aim of this study was to determine the phase of the season during which variables tested achieve their peak values. In fact, maximal values within ranges of intensity from low through to very high were obtained in the 4th phase, whereas lowest values were in the 6th phase. Players’ achievement of highest values for match running performance in the 4th phase of the season may reflect a winter break between rounds [42,46]. Players inter alia use this to regenerate their bodies and maintain exercise capacity through individual training. In addition, players participate in a few days’ preparatory camp for the spring round of matches, with the goal of increasing their multidimensional capacity. However, despite players in the 6th phase of the season maintaining match running performance similar to that observed in the 1st phase, significant decreases between the 4th and 6th phases were found. Considering that distances covered at intensity ranges from 11–13.99 to 21–23.99 km·h^−1^ in the 5th phase were reduced and that the lowest values recorded were in the 6th and last phase of the season. The suggestion is that the training stimulus obtained during the winter break, along with the different training stimuli repeated periodically through a season, are not sufficient to allow exercise capacity to be retained through to the end of the season. Our results show that, for the players competing in most full matches in the season, the 4th phase represented a breakthrough point regarding the maintenance of their physical capacities. 

Results from previous studies have shown that, when the time-delay between soccer matches is short, residual fatigue accumulated over successive matches and subsequent incomplete recovery can affect ensuing physical performance [11,12,39]. Increasing fatigue and tiredness in the 5th and 6th phases of the season intensify the burden of playing additional matches in national cup tournaments, the Champions League or Europa League. Moreover, many top players are also employed in matches played by their national teams during the season [1,9]. Coaches, sports scientists and medical teams should consider a higher frequency of player rotation, especially in the 5th and 6th phases of the season, in order that the maintenance of health and performance can be facilitated [1].

Identifiable limitations on this study reflect the authors’ failure to account for various playing positions. Future studies would also benefit from an examination of how play at different times of the day (afternoon vs evening) may make a difference, along with the weather at the time games were played (summer vs winter matches), and its impact on physical and technical activity during elite soccer matches. Furthermore, it would be interesting to check match outcomes (i.e., whether the team was winning, losing or drawing) and players playing different numbers of matches (i.e., only playing in the domestic league or playing in the domestic league plus the European cups).

## 5. Conclusions

The findings from the present study are noteworthy in that they are the first to highlight the situation of professional soccer players playing the most matches during a season. Summarising the longitudinal studies, we can suggest that practitioners need a good understanding of players’ fatigue responses to both training and matches, and of how subsequent training loads may be managed effectively to facilitate players’ preparation for the different parts of the season. 

It was shown that as they reduce the distance covered at intensities below 11 km·h^−1^, players may cover longer distances at intensities in the range from low to very high up to the 4th phase of the season. Moreover, it has been demonstrated that players are able to maintain levels of both match running and technical performance between the first and sixth phase of the season. However, despite players preparing for the season in summer, both the 1st phase and the 6th phase of the season represent periods in which the variables studied assumed similar, lowest, values. The highest values for match running performance are in turn achieved by players in the season’s 4th phase, while the lowest are those characterising the 6th phase. In the case of the Bundesliga, the 4th phase of the season represents a breakthrough point when it comes to exercise capacity among participating players being maintained. The research shows that coaches and training staff should pay more attention to capacity, with power aerobic training required pre-season, in the break between rounds and in the final phase of the season (with a view to players’ reduced physical activity being counteracted).

## Figures and Tables

**Figure 1 ijerph-16-02044-f001:**
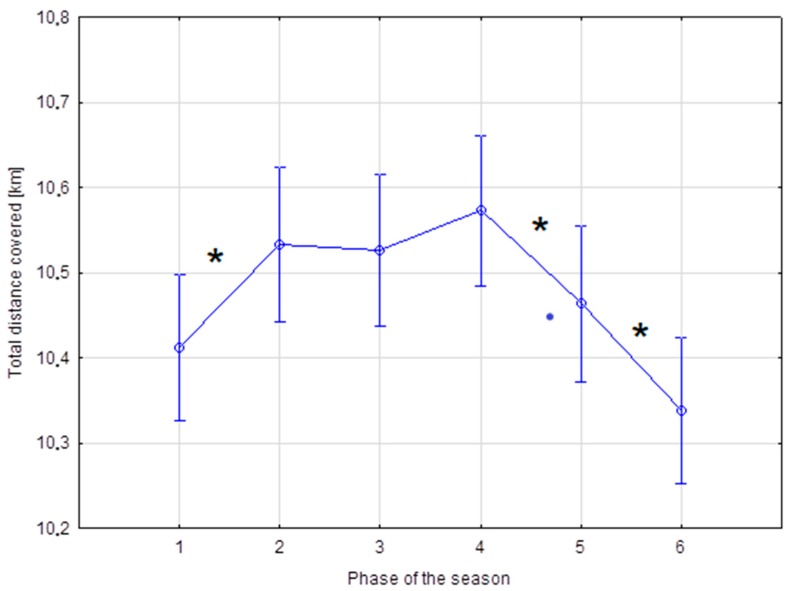
Total distance covered [km] in successive phases of the Bundesliga season. * Statistically significant difference (*p* ≤ 0.05).

**Table 1 ijerph-16-02044-t001:** Differences in [km] distances covered at different intensities by soccer players in the Bundesliga, with regard to phases of the season (mean ± SD).

Intensity Range	Phases of the Season						F (Sig.)	Statistically Significant Difference (*p* ≤ 0.05)
	1st	2nd	3rd	4th	5th	6th		
< 11 km·h^−1^	6.36 ± 0.23	6.30 ± 0.27	6.24 ± 0.26	6.19 ± 0.27	6.20 ± 0.27	6.25 ± 0.27	21.27 (0.001)	1st > 2nd; 2nd > 3rd; 3rd > 4th; 5th < 6th
11–13.99 km·h^−1^	1.73 ± 0.29	1.78 ± 0.31	1.79 ± 0.31	1.82 ± 0.29	1.78 ± 0.28	1.73 ± 0.28	10.90 (0.001)	1st < 2nd; 4th > 5th; 5th > 6th
14–16.99 km·h^−1^	1.06 ± 0.27	1.12 ± 0.31	1.13 ± 0.29	1.18 ± 0.31	1.13 ± 0.31	1.08 ± 0.29	18.29 (0.001)	1st < 2nd; 3rd < 4th; 4th > 5th; 5th > 6th
17–20.99 km·h^−1^	0.76 ± 0.20	0.80 ± 0.22	0.82 ± 0.22	0.85 ± 0.24	0.82 ± 0.24	0.77 ± 0.24	18.12 (0.001)	1st < 2nd; 2nd < 3rd; 3rd < 4th; 4th > 5th; 5th > 6th
21–23.99 km·h^−1^	0.29 ± 0.09	0.30 ± 0.09	0.31 ± 0.10	0.31 ± 0.10	0.31 ± 0.10	0.29 ± 0.10	6.28 (0.001)	5th > 6th
> 24 km·h^−1^	0.22 ± 0.10	0.23 ± 0.10	0.23 ± 0.10	0.23 ± 0.10	0.24 ± 0.10	0.22 ± 0.09	1.98 (0.021)	5th < 6th

**Table 2 ijerph-16-02044-t002:** Differences in levels of technical performance achieved by soccer players in the Bundesliga, with regard to phases of the season (mean ± SD).

Variables	Phases of the Season						F (Sig.)	Statistically Significant Difference (*p* ≤ 0.05)
	1st	2nd	3rd	4th	5th	6th		
Ball possession [s]	58.31 ± 12.98	61.38 ± 14.33	62.01 ± 13.74	61.50 ± 13.41	60.05 ± 14.11	60.05 ± 13.40	2.07 (0.028)	1st < 2nd
Passes [number]	40.13 ± 13.21	43.29 ± 14.07	43.76 ± 14.48	42.94 ± 12.88	41.92 ± 13.70	42.05 ± 14.00	2.08 (0.027)	1st < 2nd
Pass accuracy [%]	74.84 ± 9.32	75.21 ± 7.84	74.72 ± 8.47	73.82 ± 9.01	75.63 ± 8.04	74.49 ± 9.13	1.01 (0.412)	-
1-on-1 duels [number]	16.71 ± 4.80	16.82 ± 4.34	17.24 ± 4.56	17.70 ± 5.22	17.16 ± 5.42	16.20 ± 4.85	2.37 (0.079)	-
Wins of 1-on-1 duels [number]	9.32 ± 2.61	9.23 ± 2.36	9.54 ± 2.28	9.92 ± 2.66	9.53 ± 3.04	9.03 ± 2.36	1.96 (0.084)	-
Wins of 1-on-1 duels [%]	56.60 ± 10.12	55.86 ± 9.57	55.72 ± 8.29	56.99 ± 8.29	55.79 ± 8.83	56.73 ± 7.90	0.55 (0.735)	-

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
