# Peer review of "Players’ Physical Performance Decreased After Two-Thirds of the Season: Results of 3 Consecutive Seasons in the German First Bundesliga"

_ijerph, 2019, doi:10.3390/ijerph16112044_

Round 1

Reviewer 1 Report

Variability of physical and technical performance among players between matches is likely to be influenced by a myriad of factors. Hence  any monitoring of players running and of technical performance should be conducted over an extended period of time to provide comprehensive estimation of the variability. The study addressed an issue which can directly impact practice settings. The data received from professional soccer during six consecutive phases of three competitive seasons in the German first Bundesliga are unique. However, some detailed information should be added to make the study easier to compare with the findings of other researchers. The authors omitted important information (e.g. the length of each phase or why they chose six consecutive phases). The data were collected from 70 out of 556 outfield players who played at least 80% of the matches in each phase of the season. It is not clear whether players completed entire matches or played only a part of a given match. I recommend expressing such information in minute time spent on the pitch.   

Also, information about the number of players from different outfield positions should be added (e.g. central defender, wide defender, central midfield, attacker, etc.) to determine the influence of playing postions on variability of physical and technical performance.

The obtained results are in general in line with the results of other authors. What is worth noting is that no significant changes in technical activity through the season were reported.

Moreover, in Discussion, lines 192-193: 'Our results can further inform off-season individual training or modification of the training load in the summer pre-season, especially in terms of aerobic endurance' - it is too speculative. The authors didn't examine the content of off-season individual training or training load.

Lines 230-231: '...the training stimulus obtained during the winter break only sufficed for the first 6 matches of the spring round'. - in the paper there isn't strong evidence to make such a speculation. Different training stimuli are repeated periodically during the whole season.

Conclusions are consistent with the results of the study.

References - line 311: should be Chmura, P.

To improve the paper, some information about limitations of the study should be included.

Author Response

Dear Reviewer,

In response to your comments in reviewing the manuscript “Players’ physical performance decreased after two-thirds of the season: Results of 3 consecutive seasons in the German first Bundesliga”, we would like to inform you of corrections made and augmentations provided.

Variability of physical and technical performance among players between matches is likely to be influenced by a myriad of factors. Hence  any monitoring of players running and of technical performance should be conducted over an extended period of time to provide comprehensive estimation of the variability. The study addressed an issue which can directly impact practice settings. The data received from professional soccer during six consecutive phases of three competitive seasons in the German first Bundesliga are unique. However, some detailed information should be added to make the study easier to compare with the findings of other researchers. The authors omitted important information (e.g. the length of each phase or why they chose six consecutive phases). The data were collected from 70 out of 556 outfield players who played at least 80% of the matches in each phase of the season. It is not clear whether players completed entire matches or played only a part of a given match. I recommend expressing such information in minute time spent on the pitch.   

Thank you for your valuable suggestions. The intention of the work was to divide each round into three parts to ensure the thorough analysis of the beginning, middle and end of the autumn and spring round. This was to be a reflection of the number of matches, irrespective of the duration of the given phase. This has been presented in the experimental approach. (Responding to the Reviewer's question: the length of each phase was 6 weeks. In phases 2 and 5 the players had 5 Bundesliga matchdays and a one-week break for national teams). The authors added information to the text concerning the fact that the players qualified for analysis played whole matches.

Also, information about the number of players from different outfield positions should be added (e.g. central defender, wide defender, central midfield, attacker, etc.) to determine the influence of playing postions on variability of physical and technical performance.

Thank you also for this valuable suggestion. Our research focused on the problem of changes in physical and technical activity of players in subsequent stages of the season. We took account of the players who played in each phase of the three seasons in the Bundesliga at least 80% of the matches, and furthermore played whole matches.

The position distribution is as follows: Central defenders n=32; Full-backs n=22; Central midfielders n=10, Forwards n=6. As can be seen from our analysis, there were no wide midfielders, due to the fact that it is this position that is characterised by the highest frequency of change. In our opinion, the complete absence of one position from the data disqualifies the factor of position on the pitch from being taken into account. This fact was mentioned among the limitations detailed at the end of the discussion and it represents a contribution to further research in this field.

The obtained results are in general in line with the results of other authors. What is worth noting is that no significant changes in technical activity through the season were reported.

The research shows that, on 34 match days in the Bundesliga, only physical activity is modified, while technical activity remains relatively stable. These relationships may be subject to further research.

Moreover, in Discussion, lines 192-193: 'Our results can further inform off-season individual training or modification of the training load in the summer pre-season, especially in terms of aerobic endurance' - it is too speculative. The authors didn't examine the content of off-season individual training or training load.

Drawing on research (Carling et al. 2015, Bush et al. 2015, Paul et al. 2016 and others) that equates distances covered with the general aerobic preparation of players for a season, we allowed ourselves in the Discussion Section to comment on aerobic preparation. This reflects the way the latter can affect changes of this variable at least indirectly. Values are significantly lower at the start of the season (1st vs 2nd phase), which may implicate coaches and training staff in offering insufficient amounts of aerobic capacity and power training stimuli during the pre-season.

Lines 230-231: '...the training stimulus obtained during the winter break only sufficed for the first 6 matches of the spring round'. - in the paper there isn't strong evidence to make such a speculation. Different training stimuli are repeated periodically during the whole season. 

Thank you for your this. It was modified in line with the Reviewer's suggestion, with regard had to periodic training stimuli across a whole season.

Conclusions are consistent with the results of the study.

We are grateful for this positive opinion.

References - line 311: should be Chmura, P.

The correction suggested by the Reviewer was made in the text.

To improve the paper, some information about limitations of the study should be included.

At the end of the Discussion, a short paragraph on limitations, as well as further directions of study, was added.

We would like to express our sincere thanks to the Reviewer for time spent and valuable comments on our study offered. We hope that the corrections and revisions will improve the editorial value and content of our study, and allow us have our work published in your renowned Journal.

Reviewer 2 Report

The purpose of the research can not be "examine".

Information about players is insufficient.

"Conclusions" are not conclusions - there are summary.

Author Response

Dear Reviewer,

In response to your comments in the review of the manuscript “Players’ physical performance decreased after two-thirds of the season: Results of 3 consecutive seasons in the German first Bundesliga”, we would like to inform you of corrections and augmentations made.

The purpose of the research cannot be "examine".

The authors changed the word from examine to investigate.

Information about players is insufficient.

Thank you for your valuable suggestion. The authors added information to the effect that only players who played whole matches were analysed. Information about the weights and heights of players can be found in the section on players and match data.

We did not take account of player-position information because the position distribution is as follows:  Central defenders n=32; Full-backs n=22; Central midfielders n=10, Forwards n=6. As can be seen from this, there were no wide midfielders, in line with the way that this position experiences the highest frequency of change. In our view, the lack of wide midfielders precludes account being taken of this factor. Hence no analysis for position taken on the pitch was carried out. This fact was mentioned in the section on limitations present at the end of the Discussion, and does represent a contribution to further research in this field.

"Conclusions" are not conclusions - there are summary.

In line with the suggestion from the Reviewer, changes were introduced into the Conclusions section.

We would like to express our sincere thanks to the Reviewer for time spent and valuable comments on our study made. We hope that the corrections and revisions described here will improve the editorial value and content of our study, and allow us have our work published in your renowned Journal.

Reviewer 3 Report

Firstly, I would like to commend the authors on the manuscript, I really enjoyed reviewing it. Please find the following comments and suggestions for the manuscript.

Introduction

I feel the introduction is a real strength of the manuscript. Clearly explains the need for the research and the gap that it will fill within the literature. Also discusses the implications the results could have for professional sports club well. 

Specific comments

Lines 50-52: When listing causes of fatigue, should also consider psychological fatigue.

Line 64: After the comma, this part of the sentence may need to be re-worded. Not sure if it makes sense…

Line 89-90: Hypothesis 1, did you only hypothesise that it would change? Or was there a particular direction of change that you expected (e.g. Decrease in performance as season went on)? Maybe consider adding to this hypothesis (if you in fact hypothesized a change in a particular direction).

Methods

Did you consider player position? Whether a player was a defender, midfield or forward could influence the amount of touches or running data. I foresee an issue (very small) when a player was given the arithmetic mean for missing a game, especially if there is differences in the data dependent on player position. I feel this should be mentioned as a potential limitation. Furthermore, could also mention that focusing on player position, could be an area for future research J

Specific comments

Line 103 – Could the mean age, height mass etc. be put in brackets after mentioning the amount of participants? That way you don’t have to mention it later.

Line 121-123: I feel you need to mention the reliability and validity statistics of these studies. Rather than just stating, “…the reliability and validity has been assessed elsewhere.” Provide some stats of how reliable/ valid it is.

Results

Results section is very clear and concise. Reads very well.

Discussion

The discussion section is the one area that may need some improvement. As a whole, I feel you explain the implication of the study really well in the introduction, however, this doesn't come across as strong in the discussion section. There needs to be more emphasis on what the implications of the results are for conditioning / high performance staff (in my opinion).

Specific comments

Line 175: Need to keep this consistent throughout. At times, you report the velocity bands in terms of actual km/h, whilst on other occasions you use the name you have given for each velocity band. I think this would read better if you consistently refer to each velocity band with the name you have provided.

I feel there are a few confounding variables, that at least need to be mentioned as limitations of the study. Were all the games throughout the season played at different times of the day/ conditions? Although, this cannot be controlled, should be mentioned as a limitation.

The 34 game Bundesliga season is broken down into six phases. However, in the introduction, the authors discuss that the European soccer season can include up to 60 matches (due to other competitions). I feel the author needs to discuss the demands of other competitions and in what phase most of these games are played. For example, is there more of these extra games in a particular phase? Could these other games effect the results of a particular phase in the Bundesliga? I think this could be a discussion point in the discussion section, but at the very least, should be mentioned as a limitation (if it was not considered)

Author Response

Dear Reviewer,

In response to your comments in the review of the manuscript “Players’ physical performance decreased after two-thirds of the season: Results of 3 consecutive seasons in the German first Bundesliga”, we would like to inform you about corrections and augmentations made.

Firstly, I would like to commend the authors on the manuscript, I really enjoyed reviewing it. Please find the following comments and suggestions for the manuscript.

Introduction

I feel the introduction is a real strength of the manuscript. Clearly explains the need for the research and the gap that it will fill within the literature. Also discusses the implications the results could have for professional sports club well. 

Thank you for this positive opinion.

Specific comments

Lines 50-52: When listing causes of fatigue, should also consider psychological fatigue.

The authors have added information on this to the text.

Line 64: After the comma, this part of the sentence may need to be re-worded. Not sure if it makes sense…

For a better understanding of this sentence after the comma, the text was removed.

Line 89-90: Hypothesis 1, did you only hypothesise that it would change? Or was there a particular direction of change that you expected (e.g. Decrease in performance as season went on)? Maybe consider adding to this hypothesis (if you in fact hypothesized a change in a particular direction).

In the hypothesis, we assumed that subsequent stages of the season would be characterised by significant changes in physical and technical activity. The most likely nature of this change would be decreased activity due to growing residual fatigue as a result of play in further consecutive matches of the season.

Methods

Did you consider player position? Whether a player was a defender, midfield or forward could influence the amount of touches or running data. I foresee an issue (very small) when a player was given the arithmetic mean for missing a game, especially if there is differences in the data dependent on player position. I feel this should be mentioned as a potential limitation. Furthermore, could also mention that focusing on player position, could be an area for future research J

Thank you for your valuable suggestion. The position distribution is as follows: Central defenders n=32; Full-backs n=22; Central midfielders n=10, Forwards n=6. As can be seen from that analysis, there were no wide midfielders, in line with the way in which this position is affected by the highest frequency of changes. Our view was that the absence of information on wide midfielders precluded any account being taken of the factor of position on the pitch. This fact was mentioned in the context of information on limitations of the study which is present at the end of the Discussion and does represent a contribution to further research in this field.

Specific comments

Line 103 – Could the mean age, height mass etc. be put in brackets after mentioning the amount of participants? That way you don’t have to mention it later.

The correction suggested by the Reviewer has been made in the text.

Line 121-123: I feel you need to mention the reliability and validity statistics of these studies. Rather than just stating, “…the reliability and validity has been assessed elsewhere.” Provide some stats of how reliable/ valid it is.

The correction suggested by the Reviewer has been made in the text.

Results

Results section is very clear and concise. Reads very well.

We are grateful for this positive opinion.

Discussion

The discussion section is the one area that may need some improvement. As a whole, I feel you explain the implication of the study really well in the introduction, however, this doesn't come across as strong in the discussion section. There needs to be more emphasis on what the implications of the results are for conditioning / high performance staff (in my opinion).

Thank you for your valuable suggestion. A last paragraph on limitations and further testing has been added, along with practical implications (at the end of the Conclusions).

Specific comments

Line 175: Need to keep this consistent throughout. At times, you report the velocity bands in terms of actual km/h, whilst on other occasions you use the name you have given for each velocity band. I think this would read better if you consistently refer to each velocity band with the name you have provided.

In the Section on data collection and analyses, the authors provide individual speed ranges and  corresponding names for the intensity of running taking place. This accounts for interchangeability of use in the text.

I feel there are a few confounding variables, that at least need to be mentioned as limitations of the study. Were all the games throughout the season played at different times of the day/ conditions? Although, this cannot be controlled, should be mentioned as a limitation.

Thank you for this. Regardless of month, all matches in the season were played from one of the standard kickoff times in the afternoon or evening (i.e. 15.30, 18.30 or 20.30 PM). Reference to this confounding variable was made in the last paragraph of the Discussion, relating to limitations on this study and further research.

The 34 game Bundesliga season is broken down into six phases. However, in the introduction, the authors discuss that the European soccer season can include up to 60 matches (due to other competitions). I feel the author needs to discuss the demands of other competitions and in what phase most of these games are played. For example, is there more of these extra games in a particular phase? Could these other games effect the results of a particular phase in the Bundesliga? I think this could be a discussion point in the discussion section, but at the very least, should be mentioned as a limitation (if it was not considered)

Thank you for your valuable suggestion. The intention of the work was to divide each round into three parts to ensure thorough analysis of the beginning, middle and end of the autumn and spring round, in line with the number of matches, and regardless of the duration of the given phase. This information has been presented under the experimental approach. (The length of each phase was 6 weeks. In phases 2 and 5, players had 5 Bundesliga match days, as well as a one-week break for national teams).

Changes in physical and technical activity were only analysed in respect of the German first Bundesliga (34th round). We did not therefore take any account of additional matches played by certain teams in European and national cup competitions. This information was added to the last paragraph in the Discussion, as it concerns limitations and future research.

We are currently working on adding more seasons, and will therefore take the above idea into account. We will divide the players into those who played only in the domestic league, and those who played in the domestic league and in European cups – given that the number of matches played by the players in the season will definitely be greater.

We would like to express our sincere thanks to the Reviewer for time spent, and valuable comments on our study supplied. We hope that the corrections and revisions will improve the editorial value and content of our study, and allow us have our work published in your renowned Journal.